# Bamboo Plant Part Preference Affects the Nutrients Digestibility and Intestinal Microbiota of Geriatric Giant Pandas

**DOI:** 10.3390/ani13050844

**Published:** 2023-02-25

**Authors:** Ying Yao, Wenjia Zhao, Guilin Xiang, Ruiqing Lv, Yanpeng Dong, Honglin Yan, Mingxi Li

**Affiliations:** 1Chengdu Research Base of Giant Panda Breeding, Sichuan Key Laboratory of Conservation Biology for Endangered Wildlife, Chengdu 610081, China; 2School of Life Science and Engineering, Southwest University of Science and Technology, Mianyang 621010, China

**Keywords:** giant panda, bamboo part, aging, microbiome, nutrient digestibility

## Abstract

**Simple Summary:**

Bamboo part preference and a panda’s age have been shown to shift the gut microbiota composition of the giant panda, thus eliciting changes in their nutrient utilization capacity. The present study compared the differences in nutrient digestibility and fecal microbiota composition between adult and geriatric captive giant pandas when fed exclusively with a diet comprising of either bamboo shoots or leaves. Bamboo part preference exerted a significant effect on nutrient digestibility and fecal microbiota composition in both adult and aged giant pandas. Bamboo part dominated over age in shaping the nutrient digestibility and gut microbiota composition of giant pandas.

**Abstract:**

Bamboo part preference plays a critical role in influencing the nutrient utilization and gastrointestinal microbiota composition of captive giant pandas. However, the effects of bamboo part consumption on the nutrient digestibility and gut microbiome of geriatric giant pandas remain unknown. A total of 11 adult and 11 aged captive giant pandas were provided with bamboo shoots or bamboo leaves in the respective single-bamboo-part consumption period, and the nutrient digestibility and fecal microbiota of both adult and aged giant pandas in each period were evaluated. Bamboo shoot ingestion increased the crude protein digestibility and decreased the crude fiber digestibility of both age groups. The fecal microbiome of the bamboo shoot-fed giant pandas exhibited greater alpha diversity indices and significantly different beta diversity index than the bamboo leaf-fed counterparts regardless of age. Bamboo shoot feeding significantly changed the relative abundance of predominant taxa at both phylum and genus levels in adult and geriatric giant pandas. Bamboo shoot-enriched genera were positively correlated with crude protein digestibility and negatively correlated with crude fiber digestibility. Taken together, these results suggest that bamboo part consumption dominates over age in affecting the nutrient digestibility and gut microbiota composition of giant pandas.

## 1. Introduction

The giant panda (*Ailuropoda melanoleuca*) is a highly specialized herbivorous species of ursid that consumes bamboo as the primary and almost exclusive diet. Unlike most herbivores, the giant panda has no apparent internal gastrointestinal adaptions to its bamboo-dominated diet, and exhibits a short digestive tract with a rapid passage of digesta, which is similar to the gastrointestinal tract morphology of most carnivores [1]. The extremely high amount of bamboo consumption each day and low energy expenditure can partly explain how giant pandas persist solely on bamboo, a high fibrous plant with low nutritional value and digestibility [2]. However, the giant panda has been shown to lack homologs of the enzymes needed for the degradation of structural carbohydrates, the key component of bamboo [3]. It has thus been believed that the utilization and extraction of nutrients from the bamboo diet largely depends on the gut microbiome of the giant panda, as the giant panda gut microbiome has been found to exhibit a high abundance of putative genes involved in carbohydrate degradation, suggesting high utilization potential of structural polysaccharides [1,4].

Both wild and captive pandas exhibit seasonal changes in bamboo part preference, with shoots consumed in spring and summer, leaves in autumn and winter, and culms in the transition period, namely later winter and early spring [5,6]. Dietary changes are an important factor influencing the composition and function of the gut microbiome [7]. Evidences have been accumulated to show the giant panda’s gut microbiota are shaped by the seasonally-driven shifts in bamboo part preference, as the nutrient content in different parts of bamboo varies significantly, with higher cellulose, hemicellulose, and starch, as well as lower proteins, in the leaves and culms than in shoots [3,8,9]. Gut microbiota has been shown to significantly affect the nutrient utilization capacity and health status of the host [10]. In captive giant pandas, the apparent digestibility of bamboo parts differed significantly, resulting in different degrees of nutrient retention used by gut microbes in the hindgut [8]. Therefore, the changes in gut microbiome elicited by different bamboo part consumption would significantly affect the nutrient digestibility of the giant pandas.

Aging is an inevitable biological process in an organism that leads to an increased risk of many diseases [11]. In terms of longevity, captive giant pandas generally have a lifespan of almost 30 years, and individuals older than 20 are considered to be “geriatric” because the reproduction process of the giant panda generally ends after this age [12]. Aging has been proven to significantly shape the structure of gut microbiota and affect the immune and metabolic functions of giant pandas [13]. Likewise, impaired digestive function and higher risk of gastrointestinal disorders have been recognized in aged giant pandas [12]. The seasonal variation in bamboo part consumption has been shown to significantly affect the nutrient digestibility of captive giant pandas [6]. However, little is known about the effects of bamboo part preference on aged giant pandas, especially the changes of gut microbiome and nutrients digestibility. To address this issue, the nutrients digestibility and gut microbiota composition were compared between adult and older captive giant pandas when fed exclusively with a diet comprising of either shoots or leaves.

## 2. Materials and Methods

### 2.1. Ethics Statement

All protocols for the present study that involved animal care and treatment were approved by the Institutional Animal Care and Use Committee of Chengdu Research Base of Giant Panda Breeding (No. 2020010).

### 2.2. Study Subjects and Animal Husbandry

A total of 11 adult (aged 9–17 years, average age was 13) and 11 geriatric (aged 20–37 years, average age was 25) captive giant pandas were the subjects of the present study. All subjects were singly housed at the Chengdu Research Base of Giant Panda Breeding (CRBGPB, Chengdu, Sichuan, China), and all were considered healthy and were not under any medical treatment during the study period. The ambient temperature was maintained at 15 °C–22 °C, and the air humidity was 65–75%. All giant pandas were fed according to the normal husbandry practices of the CRBGPB as described in Wang et al. [6]. Bamboo was provided to giant pandas three times each day (08:00, 14:00, and 20:00). In the present study, giant pandas were given free access to bamboo and water, and the specific bamboo part was offered according to the seasonal shifts. In CRBGPB, bamboo shoots of *Phyllostachys nidularia Munro* were consumed by pandas in autumn and bamboo leaves of *Bashania fargesii* were provided to pandas in winter. In addition to the supply of bamboo parts, dietary supplements were provided daily and of the same mass to all subjects. In this study, both adult and geriatric pandas were provided with bamboo shoots for 3 months and bamboo leaves for 3 months: bamboo shoot-fed adult (AS), bamboo leaf-fed adult (AL), bamboo shoot-fed old (OS), and bamboo leaf-fed old (OL) giant pandas.

### 2.3. Sample Collection

At the last day of each period during which pandas were offered the corresponding bamboo part, fecal samples were collected from each giant panda. For each panda, the spontaneous excreted fecal samples were collected within 10 min of defecation after the feeding in the morning. To avoid contamination, samples were collected only after the floor was cleaned and disinfected. Furthermore, the outer layer of feces that contacted the floor was discarded and only fecal parts that did not touch the floor were kept and stored at −80 °C pending further analysis.

### 2.4. Apparent Nutrient Digestibility Measurement

During the last three days of each single-bamboo-part consumption period, the apparent nutrient digestibility of the corresponding bamboo part was determined in both adult and older giant pandas. The amount of ingested food and excreted feces of each individual giant panda was weighed. The bamboo samples that pandas consumed and fecal samples were collected twice a day, weighed, and immediately stored at 4 °C. During the next day, corresponding proportions of fecal samples were kept and mixed according to the amount of daily excreted feces. Finally, about 1 kg of bamboo leaves and 1.5 kg of the corresponding fecal samples, as well as 5 kg of bamboo shoots and the corresponding fecal samples, were kept at −80 °C for long-term storage. The bamboo and fecal samples were dried, ground, and sieved through a 0.45 mm sieve, then mixed, sampled, and stored at −20 °C. The chemical components of the bamboo and fecal samples were determined according to the AOAC analysis method [14]. An oven drying method was adopted to measure the dry matter (DM) content, the Kjeldahl method was used to determine the crude protein (CP) content, the Soxhlet extraction method was applied to evaluate the ether extract (EE) content, the continuous extraction of samples by dilute acids and bases was used to measure crude fiber (CF), and lastly, the oxygen bomb calorimeter calorimetric method was used to analyze the gross energy (GE) concentration of bamboo and fecal samples.

The calculation equation of apparent nutrient digestibility was as follows:$$\mathrm{Apparent}\;\mathrm{digestibility}=\frac{{\;\mathrm{Daily}\;\mathrm{intake}\;\times \;\mathrm{Nutrient}\;\mathrm{substance}\;}_{\left(\mathrm{Bamboo}\right)}-{\;\mathrm{Daily}\;\mathrm{feces}\;\times \;\mathrm{Nutrient}\;\mathrm{substance}\;}_{\left(\mathrm{Feces}\right)}}{{\mathrm{Daily}\;\mathrm{intake}\;\times \;\mathrm{Nutrient}\;\mathrm{substance}\;}_{\left(\mathrm{Bamboo}\right)}}$$

### 2.5. Genomic DNA Extraction from Feces and Sequencing

The genomic DNA of each fecal sample was isolated with the QIAamp Fast DNA Stool Mini Kits (Qiagen, Beijing, China) following the manufacturer’s instructions. The integrity and concentration of obtained DNA samples were assessed visually by agarose gel electrophoresis or measured using a NanoDrop ND-1000 device. Sterilized water was used as a negative control sample, and was included in the DNA isolation process, which showed no detectable PCR product. The common primers 515F and 806R were used to amplify the V4 region of the bacterial 16S rRNA gene, and the resulting PCR products were pooled and purified by using the Agencourt AMPureXP beads (Beckman Coulter, Brea, CA, USA) along with the MinElute PCR Purification Kit (Qiagen, Beijing, China). After pooling and purification, these amplicons were then used to construct Illumina libraries with the Ovation Rapid DR Multiplex System 1-96 (NuGEN, San Carlos, CA, USA). All of the sample libraries were sequenced on the Illumina MiSeq platform with a PE250 sequencing strategy (Novogene, Beijing, China). The raw data were deposited in the NCBI BioProject database with the accession number PRJNA916390.

### 2.6. Fecal Microbiota Analysis

The raw Illumina data were processed by Mothur software v1.3.6 (MI, USA) [15]. The high-quality paired-end sequences, which were obtained by removing the primer and barcode sequence, and also the low-quality reads, were assembled into tags with overlapping relationships. The library size of each sample was randomly subsampled into the minimum sequencing depth to minimize the biases caused by sequencing depth between samples. The USEARCH v7.0.1001 [16] was applied to cluster tags into OTUs based on 97% cut-off. The representative sequence of each OTU cluster was used for taxonomic classification against the Ribosomal Database Project database with RDP v2.6 [17]. The OTU abundance table and the OTU taxonomic assignment table laid out from the Mothur software were processed with R studio v3.4.1 [18] to calculate alpha diversity indexes of communities, as well as the beta diversity index and the Bray–Curtis distance [19]. The structural dissimilarity of the microbiota communities across the samples were visualized by non-metric multidimensional scaling (NMDS) analysis based on the Bray–Curtis distance matrix.

### 2.7. Statistical Analysis

For nutrient digestibility parameters, the statistical analysis was performed using SAS version 9.4 (SAS Institute Inc., Cary, NC, USA). Giant panda was considered the experimental unit for all analyses (*n* = 11 per treatment), and the results were expressed as means and SEM. The main effects of bamboo part and age, and the interaction between bamboo part and age were determined via two-way ANOVA. After transforming non-normal distributed data to approximately conform to normality by SAS software, the alpha indexes [20] including Observed species, Chao 1, Shannon and Simpson index as well as the relative abundance of top 10 phyla and top 30 genera were tested for significance with the one-way ANOVA, followed by Tukey’s test to evaluate the differences between treatments. Data were presented as mean ± SE. The intragroup statistic differences in beta diversity based on the Bray–Curtis distance were assessed using the one-way ANOSIM test with 10,000 permutations. Spearman’s correlation between the gut microbiota composition and nutrient digestibility parameters were calculated by the ggcor package within R software version 3.6.1 [18]. Only correlations with Spearman’s coefficient r > 0.5 and *p* < 0.05 were used to generate the network graph, which was visualized and manipulated by Gephi version 9.2 [21]. The differences were considered statistically significant when the *p* values were less than 0.05.

## 3. Results

### 3.1. Bamboo Part and Age Affect Apparent Nutrient Digestibility of Giant Pandas

A significant effect of age (F = 4.86, df = 1, *p* = 0.04) on the dietary gross energy utilization efficiency was observed showing that aged giant pandas had weaker energy extraction capacity from their diet compared to their younger counterparts (Table 1). There was a significant effect of bamboo part ((F = 203.23, df = 1, *p* < 0.001) for crude protein digestibility, indicating that bamboo shoot ingestion increased the crude protein digestibility of both adult and aged giant pandas (Table 1). There was a significant effect of bamboo part (F = 13.65, df = 1, *p* = 0.001) and age (F = 11.44, df = 1, *p* = 0.002) as well as a significant bamboo part × age interaction (*p* < 0.05) for ether extract digestibility (Table 1). This demonstrates that bamboo shoot feeding increased ether extract digestibility of aged rather than adult giant pandas when compared to bamboo leaf ingestion. Results indicated that bamboo shoot-fed giant pandas had lower crude fiber digestibility than bamboo leaf-fed counterparts (F = 16.06, df = 1, *p* < 0.001, Table 1).

### 3.2. Bamboo Part and Age Affect Fecal Microbial Profiles of Giant Pandas

After the pre-processing of raw reads, high-quality tags were generated from all samples ranging from 57,136 to 91,531, which were subsampled to 57,136 to avoid the bias induced by the sequencing depth between samples. A total of 3,728 OTUs were obtained by clustering these tags at a 97% similarity cutoff. The fecal microbiome of the bamboo shoot-fed giant pandas exhibited greater observed species (F = 4.65, df = 3, *p* = 0.01), Chao1 (F = 56.08, df = 3, *p* < 0.001), Shannon (F = 62.11, df = 3, *p* < 0.001), and Simpson index (F = 5.01, df = 3, *p* = 0.005) values than the bamboo leaf-fed counterparts regardless of age (Figure 1). The inter-group Bray–Curtis distance was significantly higher than the intra-group when giant pandas were fed with different bamboo parts independent of age (F = 25.49, df = 5, *p* < 0.001), otherwise there was no difference in the inter-group and intra-group Bray–Curtis distances (Figure 2A). The NMDS-based map also showed that the fecal microbiome of giant pandas could be sorted into two clusters by bamboo part consumption rather than age (Figure 2B), indicating the dominant role of bamboo part consumption in shaping the fecal microbiome of both adult and old giant pandas.

The predominant phyla in feces of AS, AL, OS, and OL pandas were *Firmicutes* and *Proteobacteria* (Figure 3A, Appendix A). Bamboo shoot feeding was found to decrease the relative abundance of *Firmicutes* and increase the relative abundance of *Proteobacteria* in adult giant pandas rather than old giant pandas compared to bamboo leaf consumption (Figure 3B). Additionally, bamboo shoot feeding increased the relative abundance of *Acidobacteriota*, *Actinobacteria*, and *Chloroflexi* as well as decreased the relative abundance of *Bacteroidetes* in both adult and old giant pandas compared to bamboo leaf feeding (Figure 3C). At the genus level, *Escherichia-Shigella* and *Clostridium_sensu_stricto_1* were the two most abundant bacteria in feces of all four groups (Figure 4A, Appendix A). The relative abundance of *Cellulosilyticum*, *Citrobacter*, *Enterococcus*, *Lactococcus*, *Pantoea*, *Ralstonia*, *Raoultella*, *Acinetobacter*, *Bradyrhizobium*, *Leuconostoc*, *Massilia*, and *Providenicia* were higher in feces of bamboo shoot-feeding giant pandas than the bamboo leaf-feeding group regardless of age (Figure 4B,C). Bamboo shoot intake was found to decrease the relative abundance of *Streptococcus*, *Lachnospiraceae_NK4A136_group*, and *Terrisporobacter* in feces of both adult and old giant pandas compared to bamboo leaf consumption (Figure 4B,C). Bamboo shoot feeding increased the relative abundance of *Helicobacter* and decreased the relative abundance of *Clostridium_sensu_stricto_1* in feces of adult giant pandas rather than the old group (Figure 4B,C). Compared to bamboo leaf consumption, the decreased abundance of *Escherichia-Shigella* and increased abundance of *Turicibacter*, *Hafnia-Obesumbacterium*, and *Weissella* were observed in bamboo shoot-fed old giant pandas rather than the adult group (Figure 4B,C).

### 3.3. The Correlation between Fecal Microbiota and Nutrient Digestibility in Giant Pandas

The genus *Streptococcus* and *Lachnospiraceae_NK4A136_group* were significantly positively correlated with crude fiber digestibility, whereas the genus *Lactococcus*, *Turicibacter*, *Raoultella*, *Citrobacter*, *Enterococcus*, *Pantoea*, *Cellulosilyticum*, *Weissella*, *Providencia*, and *Hafnia-Obesumbacterium* were significantly negatively correlated with crude fiber digestibility (*p* < 0.05, Figure 5). The genus *Streptococcus*, *Terrisporobacter*, and *Lachnospiraceae_NK4A136_group* were significantly negatively correlated with crude protein digestibility, whereas the genus *Lactococcus*, *Turicibacter*, *Raoultella*, *Citrobacter*, *Enterococcus*, *Ralstonia*, *Pantoea*, *Cellulosilyticum*, *Weissella*, *Providencia*, *Helicobacter*, *Hafnia-Obesumbacterium*, *Massilia*, *Bradyrhizobium*, *Leuconostoc*, and *Acinetobacter* were all significantly positively correlated with crude protein digestibility (*p* < 0.05, Figure 5). The genus *Providencia* was significantly positively correlated with ether extract digestibility (*p* < 0.05, Figure 5).

## 4. Discussion

Despite exhibiting a carnivore’s characteristic simple gastrointestinal tract, giant pandas acquire the majority of the required nutrients from bamboo. Because of the limited digestibility of plant cellulose by the giant panda genome, it was suggested that the gut microbiome may play a vital role in the digestion of this highly fibrous bamboo diet [22]. Seasonal dietary shifts in bamboo part selection have been observed in both wild and captive giant pandas, and have been shown to extensively shape the host microbiome [5]. The bamboo part preference during different seasons has been shown to significantly influence the nutrient digestibility of adult captive giant pandas, which is associated with changes in the gut microbiota composition [6]. Owing to the improvements in husbandry and veterinary care, the number of geriatric pandas in zoological institutions has increased in recent years. The aging process in giant pandas elicits a significant change in the gut microbiome, indicating that geriatric pandas exhibit a different gut microbiota composition than younger pandas [12]. While studies in humans and other animals have shown that there may exist an interaction between diet and aging in regulating host phenotype and shaping gut microbiota composition [23,24], such information in different bamboo part-fed geriatric and adult pandas remains unknown.

Unlike studies in other animals showing similar nutrient digestibility between adult and senior individuals [25,26], lower energy digestibility was found in aged giant pandas compared to the adults in the present study, indicating the declined energy extraction capacity from food in aging giant pandas. Giant pandas feed almost exclusively on bamboo, of which the different plant parts exhibit significantly different nutrient compositions [4]. Wang et al. [6] showed that the bamboo part exerted a significant effect on nutrient digestibility in giant pandas. Bamboo shoots consumption has been shown to increase the crude protein digestibility and decrease the crude fiber digestibility of giant pandas [6]. Consistently, higher crude protein digestibility and lower crude fiber digestibility were observed in bamboo shoot-fed adult and geriatric giant pandas compared to those fed with bamboo leaves in the present study, which might be attributed to the inhibition of crude protein utilization induced by the higher level of fiber in bamboo leaves [27]. In rodent models, the aging process was found to decrease lipid absorption through reducing the pancreatic lipase activity [28]. In this study, bamboo shoot consumption increased the ether extract digestibility in aged giant pandas rather than in adults compared to bamboo leaf feeding. This finding might be related to the lower lipase activity in the small intestine of senior giant pandas and the higher ether extract content in bamboo leaves. Compared with adults, the ether extract in bamboo leaves was too high for aged giant pandas to fully digest, resulting in the lower digestibility of ether extract in senior pandas fed with bamboo leaves than those fed with bamboo shoots [6].

Accumulated evidences have demonstrated the possible role of the gut microbiota in the regulation of nutrient harvest in humans and monogastric animals [29,30]. More typically, as the giant panda lacks enzymes for the digestion of bamboo, it has thus been suggested that the giant panda appears to have no alternative but to rely on symbiotic gut microbes to extract nutrients from its highly fibrous bamboo diet [31]. A previous study contended that dietary shifts induced changes in nutrient digestibility in captive giant pandas and were associated with the alteration of the microbiota composition [6]. Both bamboo plant part and age have been shown to play a critical role in shaping the gut microbiota profile in captive giant pandas [7,8,12], however the interaction between bamboo plant part and age on intestinal microbiota composition, as well as the relationship between the interaction-induced gut microbiota shifts and nutrient digestibility of the captive giant pandas, remains unknown. Consistent with the previous study showing a more diverse gut microbiome in bamboo shoot-fed giant pandas than their counterparts [8], we found that bamboo shoot feeding increased the observed species, Chao1, Shannon, and Simpson indexes in both adult and old giant pandas. This indicates that there is a more abundant and diverse microbiome in bamboo shoot-fed giant pandas.

Research showed that the elderly pandas exhibited lower bacterial species richness and diversity than the younger individuals [12,22]. However, in this study, the main effect of age on the alpha diversity indices of microbiome in giant pandas was not observed, which is inconsistent with findings in rodents in which the microbial composition was generally affected by age rather than diet [32]. This indicates the predominant role of dietary shifts rather than age in shaping the gut microbiota of giant pandas. The dissimilarity distance analysis in the present study also confirmed that the fecal microbiota of giant pandas could be sorted into two clusters by bamboo part independent of age. It has been demonstrated that phyla *Firmicutes* and *Proteobacteria* were the most predominant bacteria in the fecal microbiome of giant pandas [3,4]. In the present study, bamboo shoot feeding decreased the abundance of *Firmicutes* and increased the abundance of *Proteobacteria* in the adult group rather than the geriatric group compared to bamboo leaf feeding. This is contradictory with the previous finding that the relative abundance of *Proteobacteria* was the highest in the bamboo-leaf fed giant pandas [8]. However, in vivo studies in rodents revealed that bamboo shoot-derived components promoted the colonization of bacteria belonging to *Proteobacteria* and decreased the abundance of *Firmicutes* bacteria in the gut [33,34]. The contradictory results might stem from the different study subjects or use of different bamboo species. Previous studies in monogastric animals showed that the relative abundance of *Acidobacteriota* was positively correlated with the intake amount of dietary protein and the relative abundance of *Bacteroidetes* was negatively correlated with dietary protein level [35,36]. In the present study, the higher abundance of *Acidobacteriota* and lower abundance of *Bacteroidetes* were observed in bamboo shoot-fed giant pandas regardless of age, which might be attributed to the higher amount of protein in bamboo shoots than bamboo leaves [6]. Consistent with the previous findings [4], the genera *Escherichia-Shigella* and *Clostridium_sensu_stricto_1* were predominantly present in the fecal microbiome of giant pandas in this study. Bamboo shoot consumption has been shown to decrease the abundance of *Escherichia-Shigella* and increase the abundance of *Weissella* in the feces of giant pandas [8]. Our study further revealed that the bamboo shoot feeding-induced changes in *Escherichia-Shigella* and *Weissella* abundances were only observed in aged giant pandas. In addition, the decreased abundance of *Clostridium_sensu_stricto_1* was observed in bamboo shoot-fed adults rather than geriatric giant pandas compared to the bamboo leaf group. This finding was consistent with the previous study showing the higher abundance of *Clostridium_sensu_stricto_1* in the bamboo leaf consumption stage versus bamboo shoot consumption stage [3]. The inconsistent findings demonstrate that the genus *Clostridium_sensu_stricto* was not significantly enriched in the bamboo leaf stage and showed low sensitivity to the host’s seasonal dietary changes [1]. These contradictory results regarding the effects of bamboo part consumption on predominant genera abundance in giant pandas further suggest that the distribution of bacteria at the genus level in giant pandas might be dependent on the interaction effect of dietary shifts and age of the host. 

Seasonal variations in bamboo part selection has been shown to shape the bacteria distribution at the genus level of giant pandas [1,3]. The abundances of genera *Cellulosilyticum*, *Lactococcus*, and *Streptococcus* were significantly affected by the consumption of different bamboo parts [8]. Consistently, in this study, bamboo shoot feeding significantly increased the abundance of *Cellulosilyticum*, *Lactococcus* and other genera as well as decreased the abundance of *Streptococcus* in feces of both adult and aged giant pandas compared with bamboo leaf ingestion. In monogastric animals, the shifts in gut microbiota composition were found to closely correlate with nutrient digestibility [37]. The genus *Streptococcus* was positively related to crude fiber digestibility in pigs [38]. In this study, the genera *Streptococcus* and *Lachnospiraceae_NK4A136_group* were positively correlated with crude fiber digestibility in giant pandas, indicating the critical role of these two genera in the utilization of crude fiber of bamboo. High protein diets and ingredient consumptions have been shown to increase the abundance of the genera *Turicibacter* and *Lactococcus* in rodents [39,40]. In the present study, the genera *Turicibacter*, *Lactococcus*, and other genera were positively correlated with the crude protein digestibility of giant pandas, which indicates that these bacteria may be important for the protein utilization of the bamboo parts. Taken together, the gut microbiota composition of giant pandas was mainly shaped by bamboo part consumption rather than age.

## 5. Conclusions

In conclusion, bamboo shoot feeding increased the crude protein digestibility and decreased the crude fiber digestibility of giant pandas regardless of age. Bamboo part consumption dominated over age in shaping the gut microbiota composition of giant pandas. The shifts in taxa distribution at genus level might be responsible for the bamboo part-induced nutrient extraction alterations.

## Figures and Tables

**Figure 1 animals-13-00844-f001:**
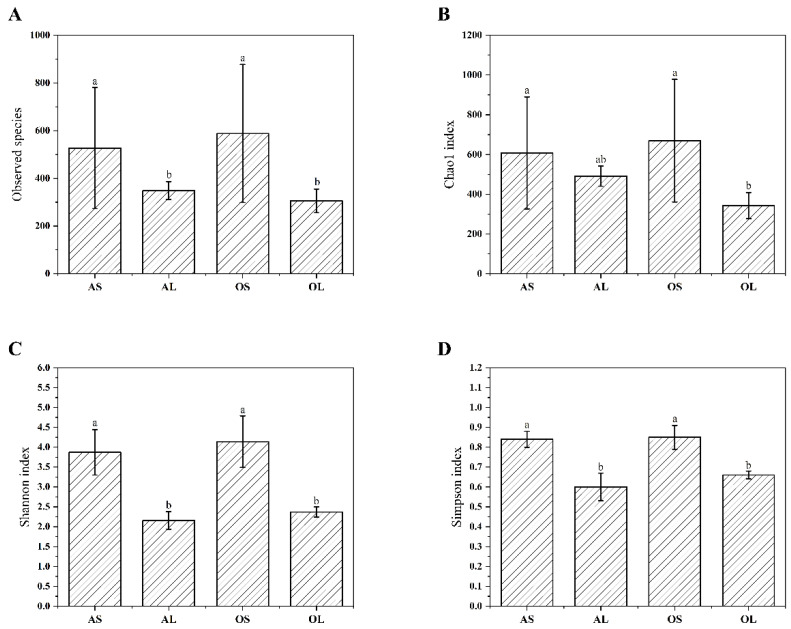
Effects of bamboo part consumption and age on alpha diversity indices of fecal microbiome in captive giant pandas. (**A**) Observed species. (**B**) Chao1 index. (**C**) Shannon index. (**D**) Simpson index. AS (*n* = 11), adult giant pandas fed with bamboo shoots; AL (*n* = 11), adult giant pandas fed with bamboo leaves; OS (*n* = 11), old giant pandas fed with bamboo shoots; OL (*n* = 11), old giant pandas fed with bamboo leaves. In each panel, values are presented as mean ± standard error; bars with different letters denote a significant difference (*p* < 0.05).

**Figure 2 animals-13-00844-f002:**
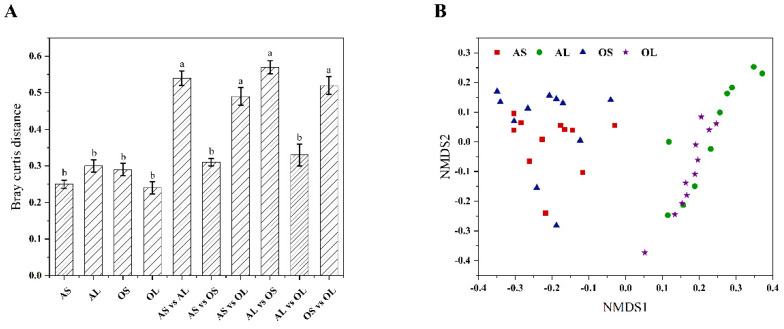
The beta diversity index and structure analysis of fecal microbiome in giant pandas. (**A**) The intra- and inter-group Bray–Curtis distance. (**B**) Non-metric multidimensional scaling (NMDS) ordination plot derived from the Bray–Curtis distances in feces of giant pandas. AS (*n* = 11), adult giant pandas fed with bamboo shoots; AL (*n* = 11), adult giant pandas fed with bamboo leaves; OS (*n* = 11), old giant pandas fed with bamboo shoots; OL (*n* = 11), old giant pandas fed with bamboo leaves. In panel (**A**), values are presented as mean ± standard error; bars with different letters denote a significant difference (*p* < 0.05).

**Figure 3 animals-13-00844-f003:**
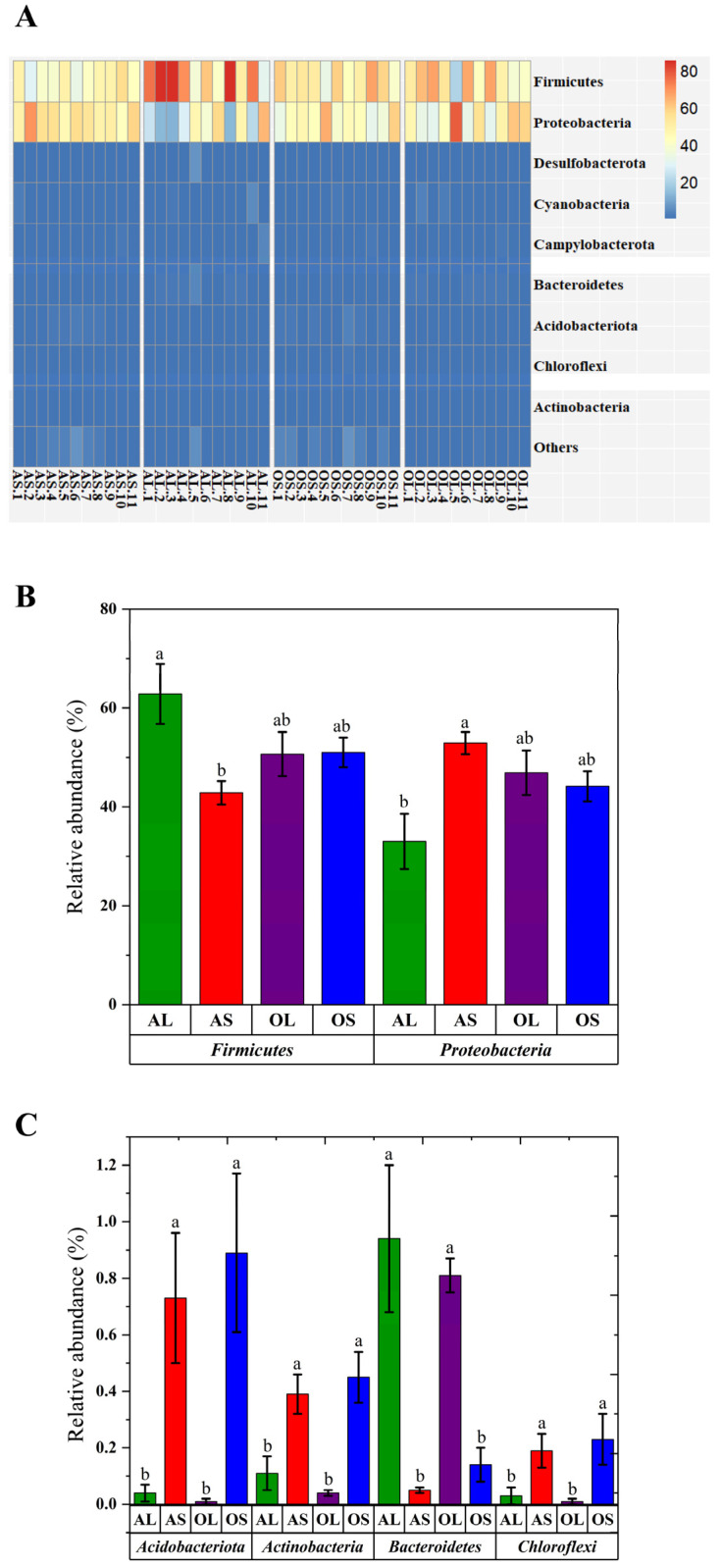
Effects of bamboo part consumption and age on the taxa distribution at the phylum level in giant pandas. (**A**) Heat map of relative abundance of phyla. (**B**,**C**) Significantly different phyla in feces among four groups. AS (*n* = 11), adult giant pandas fed with bamboo shoots; AL (*n* = 11), adult giant pandas fed with bamboo leaves; OS (*n* = 11), old giant pandas fed with bamboo shoots; OL (*n* = 11), old giant pandas fed with bamboo leaves. In panel (**B**,**C**), values are presented as mean ± standard error; bars with different letters mean significant difference (*p* < 0.05).

**Figure 4 animals-13-00844-f004:**
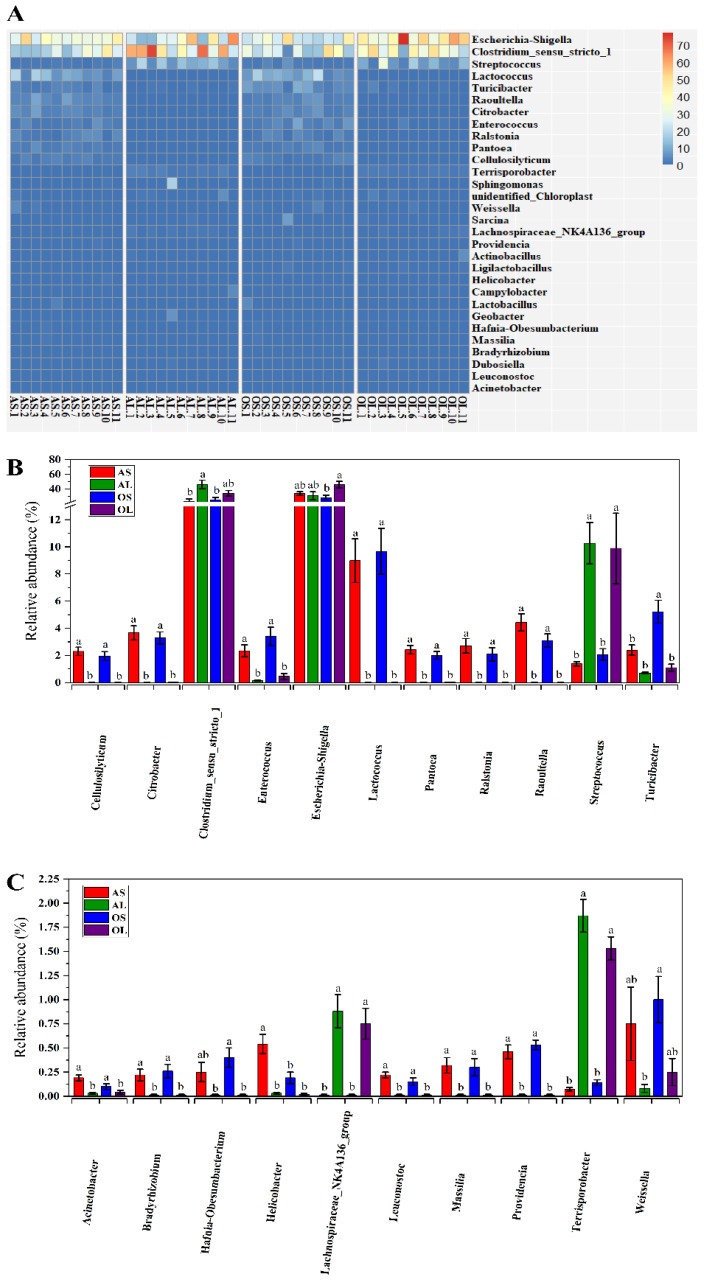
Effects of bamboo part consumption and age on the taxa distribution at the genus level in giant pandas. (**A**) Heat map of relative abundance of top 30 genera. (**B**,**C**) Significantly different genera in feces among four groups. AS (*n* = 11), adult giant pandas fed with bamboo shoots; AL (*n* = 11), adult giant pandas fed with bamboo leaves; OS (*n* = 11), old giant pandas fed with bamboo shoots; OL (*n* = 11), old giant pandas fed with bamboo leaves. In panel (**B**,**C**), values are presented as mean ± standard error; bars with different letters mean significant difference (*p* < 0.05).

**Figure 5 animals-13-00844-f005:**
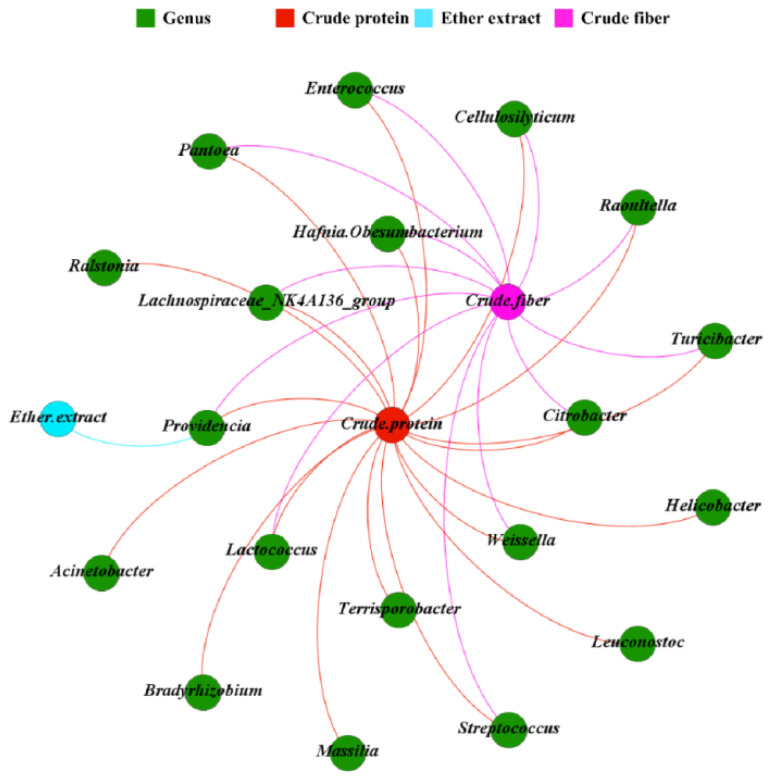
Network analysis between the top 30 genera and nutrient digestibility in giant pandas. The nodes were colored according to nutrient digestibility and genus. Only correlations with Spearman’s coefficient r > 0.5 and *p* < 0.05 are shown.

**Table 1 animals-13-00844-t001:** Effects of bamboo part and age on apparent nutrient digestibility of captive giant pandas.

Parameters	Adult Giant Panda		Old Giant Panda		SEM	*p*-Value		
	Shoots	Leaves	Shoots	Leaves		Bamboo Part	Age	Bamboo Part * Age
Dry matter (%)	26.56	32.67	32.67	26.44	3.66	0.99	0.99	0.10
Gross energy (%)	41.11 ^ab^	43.00 ^a^	35.89 ^ab^	28.11 ^b^	4.56	0.52	0.04	0.30
Crude protein (%)	81.78 ^a^	58.67 ^b^	83.11 ^a^	53.56 ^b^	1.95	<0.01	0.31	0.10
Ether extract (%)	52.56 ^a^	47.22 ^a^	49.00 ^a^	12.22 ^b^	0.77	<0.01	< 0.01	<0.01
Crude fiber (%)	14.67 ^bc^	27.67 ^a^	7.78^c^	22.01 ^ab^	4.26	<0.01	0.07	0.89

Note: Within a row, values with different letter superscripts indicate significant difference (*p* < 0.05). * means the interaction between bamboo part and individual age.

## Data Availability

The research data in this study has been deposited in the NCBI BioProject database with the accession number PRJNA916390.

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
