# Peer review of "Bamboo Plant Part Preference Affects the Nutrients Digestibility and Intestinal Microbiota of Geriatric Giant Pandas"

_animals, 2023, doi:10.3390/ani13050844_

Round 1
Reviewer 1 Report
In this manuscript, the authors studied the effects of bamboo part consumption on the nutrient digestibility and gut microbiome of adult and older captive giant pandas. Results found that bamboo part preference exerted a significant effect on nutrient digestibility and fecal microbiota composition in both adult and aged giant pandas. Furthermore, bamboo part dominated over aging in shaping the nutrient digestibility and gut microbiota composition of giant pandas. This is an innovative and logical manuscript, but there are still some problems that need to be revised.
1. One main problem is incorrect language use and grammar (singular vs. plural; past vs. present; punctuation; spaces). The correct language use is in the responsibility of the authors. You would need the assistance of a native English speaker or a professional service for language editing.
2. Line 3: change "panda " to "pandas"
3. Line 12: change " the changes" to " changes"
4. Line 26: change " significant" to " significantly"
5. Line 28: change " level" to " levels"
6. Line 41: change "panda " to "pandas". Please check the similar errors in the manuscript.
7. Line 45: change "bamboo diet " to "the bamboo diet"
8. Line 47: change "high " to " a high"
9. Line 58: change "host " to " the host"
10. Line 65: Delete " with the age"
11. Line 68: change "function" to " functions"; change " the impaired" to " impaired"
12. Study subjects and animal husbandry: Please describe the feeding and management conditions of giant pandas.
13. Line 94: change " bamboo leaves for 3 month" to " bamboo leaves for 3 months "
14. Line 97: Please describe in detail how the stool samples were collected.
15. Line 106: More detailed methods are needed to explain how you handle fecal samples for apparent nutrient digestibility determination.
16. Line 108: This sentence is not described correctly and needs to be revised. Samples should be dried, crushed, sieved, mixed and sampled.
17. Line118: Carefully check whether the formula for determining apparent digestibility is accurate.
18. Line 166: change " were" to " was "
19. Line 172: the format of Table 1 needs to be revised. “Dry matter” should not be thickened and underlined. Two decimal places are reserved for “SEM” and "P-value". Shoulder letters need to be noted. For example, “a, b, c Within a row, values with different letter superscripts mean significant difference (p < 0.05).”
20. Line 181: change " intra-group" to " the intra-group "
21. Line 185: change " dominance" to " dominant "
22. Line 198: change " heatmaps" to " heat maps"
23. Line 255-256: Please check the sentence. Does this mean the digestibility of carbohydrates?
24. Line 260: change " season" to " seasons"
25. Line 265: change " differed" to " different"
26. Line 266: change " human" to " humans"
27. Line 270: Delete " the"
28. Line 274: change " significant differed nutrient composition" to " significantly different nutrient compositions "
29. Line 278: Delete " the"
30. Line 282: change " model" to " models"
31. Line 288: change " resulting" to " resulting in"
32. Line 321: change " in gut" to " in the gut"
33. Line 324: change " positive" to " positively"
34. Line 332: change " increased" to " increase"
35. Line 341: change " sensitive" to " sensitivity"
36. Line 352: Delete " the"
37. Line 365: change " decrease" to " decreased"
Reviewer 2 Report
In the submitted manuscript (animals-2171654), the authors compare the differences in nutrient digestibility and fecal microbial community between adult and older captive giant pandas fed with shoots or leaves. Understanding of how bamboo nutrition or microbial composition influence gut microbiome of the giant pandas may be valuable in increasing the effectiveness of the captive breeding programs for giant pandas. This research is important considering the dietary impact on gut microbiome of giant panda. Importantly, results revealed bamboo part dominated in shaping the nutrient digestibility and gut microbiota composition of giant pandas. The manuscript has clear objectives and methods that are appropriate for achieving the objectives. The results are clearly presented and discussed with reference to previously published studies.
Pg 3, Figure 1. The information is already stated in the materials and methods section and is therefore redundant; please delete this Fig.
Other Comments:
1. I am not sure I understand the importance of this and how you are relating the finding of Turicibacter and Lactococcus with the crude protein digestibility. Are there any experiments can be carried out to confirm this finding in your future work? 2. Are the references appropriate and relevant? 3. Previous studies has illustrated the relationship between bamboo and the gut microbiome of giant pandas. Do you consider the topic original in the field?
Reviewer 3 Report
In this study, the authors examined the effect of bamboo part consumption on giant panda's nutrient digetibility and gut microbiome. Authors also analyzed the effects of age of giant panda on these parameters. The manuscript was well-written, just a slight modification should be needed as described below.
Minor
1. Figure legends.
In the legends of Figure 2, 3, 4 and 5, authors should explain sample size, values (mean plus/minus SEM) and what the letters shown on the bars indicated. The denoted letters of legend should be corrected to capital letters (e.g. (a) -> (A))
[Enriched Report]
1. What is the main question addressed by the research?
In this study, the authors examined the effect of bamboo part consumption on giant panda's nutrient digestibility and gut microbiome. Authors also analyzed the effects of age of giant panda on these parameters. Then authors found that the gut microbiota composition of giant pandas were affected by both factors (diet and age), however, the effect was more profound in bamboo part.
2. Do you consider the topic original or relevant in the field? Does it address a specific gap in the field?
Yes, the topic was original and relevant in the field.
3. What does it add to the subject area compared with other published material?
This article added the information about the alterations of gut microbiome according to the bamboo part, and their age to some extent.
4. What specific improvements should the authors consider regarding the methodology? What further controls should be considered?
There were no defects in the study design. Setting the independent two experimental groups is ideal, although this is too difficult in such rare animals.
5. Are the conclusions consistent with the evidence and arguments presented and do they address the main question posed?
Yes.
6. Are the references appropriate?
Yes.
7. Please include any additional comments on the tables and figures.
Figure legends.
In the legends of Figure 2, 3, 4 and 5, authors should explain sample size, values (mean plus/minus SEM) and what the letters shown on the bars indicated. The denoted letters of legend should be corrected to capital letters (e.g. (a) -> (A))
